# The Internal Capability of Vietnam Social Sciences and Humanities: A Perspective from the 2008–2019 Dataset

**Manh-Toan Ho** [1,2,*], **Thu-Trang Vuong** [3,*], **Thanh-Hang Pham** [4,5], **Anh-Phuong Luong** [2,6], **Thanh-Nhan Nguyen** [2,6] **and Quan-Hoang Vuong** [1,*]

1   Centre for Interdisciplinary Social Research, Phenikaa University, Yen Nghia Ward, Ha Dong District, Hanoi 100803, Vietnam
2   A.I. for Social Data Lab, Vuong & Associates, 3/161 Thinh Quang, Dong Da District, Hanoi 100000, Vietnam; phuongla.yrc@gmail.com (A.-P.L.); nhanthanhnguyen20@gmail.com (T.-N.N.)
3   Sciences Po Paris, 75337 Paris, France
4   Faculty of Management and Tourism, Hanoi University, Km9, Nguyen Trai Road, Thanh Xuan, Hanoi 100803, Vietnam; hangpt@hanu.edu.vn
5   School of Business, RMIT Vietnam University, Hanoi 100000, Vietnam
6   Foreign Trade University, 91 Chua Lang Street, Dong Da District, Hanoi 100000, Vietnam
*   Correspondence: toan.homanh@phenikaa-uni.edu.vn (M.-T.H.); thutrang.vuong@sciencespo.fr (T.-T.V.); hoang.vuongquan@phenikaa-uni.edu.vn (Q.-H.V.)

**Abstract:** International collaboration contributes significantly to improving scientific performance in Vietnam, but it also results in Vietnamese researchers depending on foreign partners to get their work published. The current study is an initial effort to examine the domestic research capability of Vietnam's Social Science and Humanities (SSH) disciplines through scientific productivity. The research focuses on investigating various aspects, including leading Vietnamese authors, solo authors, and gender difference, as well as international and domestic collaboration networks. The study extracts the data of 2040 Vietnamese SSH authors, 1981 foreign authors, and 3160 publications during the period of 2008–2019, from the exclusive Social Sciences and Humanities Peer Awards (SSHPA) database. Findings show a steadily rising contribution from leading domestic authors in SSH research, with an annual growth rate of approximately 22.33%. Moreover, 77.26% of publications are led by Vietnamese researchers. The proportion of publications by Vietnamese authors increased dramatically and surpassed the proportion of internationally collaborated publications in 2019, at 55.83%. The domestic research collaboration network has expanded in an observable manner. However, the participation rate of women in SSH research was relatively low, at an average of 37.30%. While the self-sustaining capacity of SSH researchers and institutes in Vietnam has been rising, gender inequality remains prevalent. In order to further strengthen and promote the scholarly community, as well as their research capacity, and to maintain public trust in SSH research, we recommend that policymakers encourage self-sustaining research, nationwide collaboration, and empower female researchers.

**Keywords:** social sciences and humanities; vietnam; self-sustaining research; domestic collaboration; women in science

## 1. Introduction

In the era of globalization, international collaboration is an inevitable trend in many areas, including science. Cooperation with researchers from other countries is shown to improve absolute scientific

output [1], and this is of critical importance for developing nations to enhance the performance and capabilities of their research communities. For example, in South and East Asian countries, international collaboration plays an important role in raising average citation impact above the global average [2]. However, dependence on international collaboration, to some extent, may also have a negative impact on the sustainability of a country's scientific productivity and quality. Therefore, building the internal capabilities of scientists in a national context is imperative for developing countries.

Vietnam, as an emerging economy, has witnessed the rapid development of science over the past decades [3]. The Global Research Report by Web of Science (2019) stated that in South and East Asia, Vietnam has the fastest growth rate, with its indexed publication volume increasing more than five-fold since 2009 and showing every sign of continuing expansion [2]. Despite this seemingly positive information, international collaboration, in fact, accounted for approximately 77% of all publications [4,5]. Noticeably, the majority of these international projects were led by foreign authors [4]. From another perspective, studies utilizing network analysis to analyze co-authorship patterns among Vietnamese researchers indicate some signals of unsustainability. These studies found that in any given scientific group, most co-authors are centered around well-connected and highly productive researchers [5], and there are some indicators of unsustainable co-authorship groups due to big gaps in productivity and connection [6].

In a regional context, among East and South-East Asian countries, Vietnam still has low research intensity primarily due to its heavy reliance on foreign co-authors and a limited quantity of international publications in applied and multidisciplinary fields [7]. During the 1991–2010 period, Vietnam's research productivity is only equivalent to 13.33% of Singapore's and 29% of Thailand's [8,9].

Among various disciplines, Social Science and Humanities (SSH) appears to hold a humble position compared with other fields in Vietnam and has been disparaged for its low productivity in terms of its total number and adjusted number, in comparison with natural science and technology disciplines [3,10]. By definition, SSH is the study of human behavior and interaction in social, cultural, environmental, economic, and political contexts [11]. Currently, SSH is not only implemented in the context of each separate discipline; instead, contemporary SSH projects are inherently interdisciplinary, taking place in different scientific and organizational environments [12]. Compared with natural sciences, SSH is more locally-oriented [13]. Therefore, in the context of Vietnam's fast-growing economy, SSH plays an important role in investigating socio-cultural as well as economic situations in order to provide reliable results for policymakers and practitioners.

Given the fact that Vietnamese authors must cooperate with international partners in scientific projects while also attempting to enhance domestic scientific sustainability, this article aims to provide a comprehensive view of the internal capabilities of the country's SSH disciplines in the period of 2008–2019. This also provides information for policymaking, as well as for social scientists to use for the advancement of science in the nation.

## 2. Literature Review

### 2.1. Lead Authors, New Authors, Solo Authors and Female Authors in Vietnam's SSH

To understand the internal capabilities of Vietnam's SSH, first we must take into account the number of solo authors and lead authors in publications. It is reported that approximately 75% of Vietnamese SSH researchers never produce any single-authored papers [14]. Meanwhile, the number of solo authors and single-authored papers worldwide are in decline, both in terms of numbers and proportions [15–17]. Also, in the context of intense collaboration with foreign partners in Vietnam's SSH, international authors are often the lead authors, which is believed to demonstrate the limited academic qualities of domestic researchers [14,18]. The findings from Vuong et al.'s (2017) study [14] provides some initial insights into the fact that Vietnamese social scientists do not often hold a leading position in collaborative projects; data shows that during a decade-long period of their career, SSH authors lead less than two scientific publications on average.

Moreover, another issue considered in this paper is the number of practicing female researchers. Globally, in various studies, female scientific productivity is reported to be the same as their male counterparts [19,20]. However, there exists an enormous gender gap regarding the number of researchers themselves: less than 30% of the world's researchers are women [21]. In Vietnam, the long-standing Confucius Society has considered academic achievements more or less a man's role, perceiving women's key responsibilities mostly in terms of housework. Until 2010, women made up only 11% of Ph.D. and Master's students, 4% of associate professors, and 1% of professors [22]. Such a low prevalence of female researchers has been attributed to gender discrimination in terms of salary, training, promotion, research funding, and committee member appointment [23]. Therefore, gender equality needs to be a crucial goal for the Vietnamese scientific community; however, few studies have examined this matter in SSH research. Among those few, Vuong et al. [17] found that there is virtually no difference between male and female contributions to Vietnamese SSH research. Therefore, encouraging more women to participate in scientific research would greatly benefit scientific performance in Vietnam by improving research productivity, flexibility, accuracy, and innovation [24,25]. In other words, gender diversity is crucial in expanding the internal capabilities of practicing science in Vietnam.

### 2.2. Collaboration Patterns of Authors in Vietnam's SSH

Regarding research output, Vietnam's research performance has achieved remarkable success in raising scientific productivity [8,26]. As an initiative to promote scientific works, the first national scientific funding agency, the National Foundation for Science and Technology (NAFOSTED), was established in 2008 by the Vietnamese government in order to provide financial support to both natural and social science fields. Together with the later issuing of Circular no. 23/2014/ND-CP, which included rigorous instructions for the organization and operation of the NAFOSTED, this has allowed progress towards making foundation-funded scientific projects compliant with international standards. Until 2019, there have been more than 10,000 participating scientists, over 2400 trained doctors, and more than 4000 Institute for Scientific Information (ISI)'s Web of Science-indexed articles published across nearly 2800 research projects, funded and supported by NAFOSTED sponsorship [27].

Partly because of these efforts, the number of the country's indexed publications has grown more than five times in the last decade [2]. According to the Global Research Report, Vietnam has 294 researchers per million population, which ranks fourth in the South and East Asia region after Singapore, Malaysia, and Thailand. The annual average publication output on Web of Science from 2014 to 2018 is 3766 papers, which ranks sixth in the region. However, profiles of papers published during the period from 2009–2018 also show that amongst more than 26,700 papers in Vietnam, 78% are conducted with international collaboration [2].

In addition, regarding research quality, the report presents a widely used indicator, namely the Category-Normalized Citation Impact (CNCI), which reflects a publication's academic impact on a country. The world average is used as a reference benchmark at 1.0. Notably, despite positive findings with regards to Vietnam's scientific output, CNCI has shown a large disparity between domestic scientific impact (CNCI domestic), which removes all publications with international co-authorship, and the gross one (CNCI gross), which counts all works with at least one national address. In particular, the CNCI gross at 1.20 is almost twice as much as the CNCI domestic at 0.67, which indicates that Vietnamese researchers have limited internal capabilities [2].

It is therefore suggested that dependency on international partnerships in terms of both quantity and quality should be reduced as such dependency may become a worrying factor in some situations. This also calls for further investigation of the internal capabilities of Vietnam's researchers in order to sustain, extend, and improve the knowledge base of the country.

In a previous work, data from a sample of 412 Vietnamese scholars who published in Scopus-indexed journals between 2008 and 2017 revealed that more than 90% of social scientists collaborated with other authors to publish articles, doing this 13 times on average [28]. The data system also showed the prevalence of international cooperation between 1996 and 2013 accounted

for 77% of the growth, which proved that Vietnamese scientific outputs depend heavily on foreign authors [4,29]. By analyzing the network of 412 authors, Ho et al. (2017) found that the network level of connection was low with only 0.47%, while the clustering factor was very high (58.64%) [30]. This demonstrated that communication and knowledge exchange between Vietnamese scholars was inefficient and socially unsustainable [31,32].

In this paper, we focus on the field of SSH because of its importance in Vietnam's rapidly changing social, cultural, environmental, economic, and political contexts. Our focus also results partly from the lack of reliable publishing data on natural sciences in Vietnam [26]. This case study, though being specific to the country, can also offer insights into global trends in general, where higher investment in science thanks to economic development may not have been utilized in the most effective and efficient way, where the goal is to contribute to the expansion of scientific knowledge. Additionally, the article aims to expand the findings of previous studies by examining the collaboration patterns of authors in Vietnam's SSH, both domestically and internationally. The questions investigated are as follows: Which countries are the most important partners of Vietnam's SSH? What proportion of publications are by Vietnamese authors only, and what proportion are the result of collaboration with international authors? Which disciplines are characterized by intensive international collaboration? How has domestic collaboration developed during the period from 2008–2019? How many higher education or research institutions in Vietnam engage in domestic cooperation?

## 3. Materials and Methods

### 3.1. Materials

This article employs the Social Sciences and Humanities Peer Awards (SSHPA, accessible online at: https://sshpa.com). It is a semi-automatic system, operating since 2017 to collect the scientific output of Vietnamese authors in SSH from 2008. The system's creation process and logical structure can be found in detail in the article by Vuong et al. (2018) [29] (The dataset is available in the Supplementary Materials).

The SSHPA system collects papers from 31 fields in SSH. Detailed information about these disciplines is listed in Table 1.

The data collection process of the SSHPA database was designed to: (i) collect data from every Vietnamese SSH researcher who has published in journals indexed in the Web of Science (WoS) or the Scopus database, or on the list of prestigious international and national sources approved by NAFOSTED [33], and (ii) ensure reliability and accuracy. Additionally, information about Vietnamese SSH researchers can only be inputted into the database if they satisfy at least two of the following criteria:

- Being affiliated with an organization in Vietnam; OR
- Publishing at least one paper about Vietnam or using data collected in Vietnam-related research in SSH disciplines.
- The SSHPA database takes into account both international and domestic perspectives on scientometrics, and the database strives to be as expansive as possible.
- For this research, the data collection process includes three steps: searching for authors, updating their profile, and assuring data quality.
- Step 1: searching for authors. Collecting information about Vietnamese social scientists who have international publications on reliable mainstream sources and cross-validating among data sources. Experts' opinions are consulted for suggestions and confirmation of new researchers.
- Step 2: updating their profile. After the data are gathered and cross-checked to ensure reliability, author information is entered into SSHPA's system.
- Step 3: assuring data quality. We check the data both manually and automatically. The data collectors check the accuracy of the information with different sources. Also, the SSHPA system can filter out duplicate and suspicious data automatically.
- More details about the data collection process are illustrated in the article by Vuong et al. (2018) [29].

**Table 1.** Disciplines in Social Sciences and Humanities (SSH) in Vietnam (2008–2019).

| Field | Number of Articles | Number of Authors |
|---|---|---|
| Economics | 941 | 1350 |
| Education | 484 | 534 |
| Health Care | 451 | 967 |
| Business | 362 | 580 |
| Sociology | 213 | 435 |
| Environment/Sustainability Science | 202 | 538 |
| Management | 154 | 269 |
| Agriculture | 143 | 326 |
| Law | 95 | 79 |
| Tourism | 90 | 121 |
| Political Science | 84 | 115 |
| Psychology | 63 | 189 |
| Media/Journalism | 40 | 53 |
| Linguistics | 39 | 33 |
| International Relations | 38 | 31 |
| Philosophy | 30 | 74 |
| Cultural Studies | 30 | 66 |
| Urban Studies | 27 | 55 |
| Anthropology | 24 | 44 |
| Applied Math | 23 | 39 |
| Scientometrics | 18 | 43 |
| Geography | 18 | 57 |
| Forestry | 15 | 33 |
| Demography | 15 | 32 |
| History | 13 | 21 |
| Art | 11 | 24 |
| Logistics | 10 | 18 |
| Asian Studies | 7 | 16 |
| Archeology | 7 | 36 |
| Literature | 6 | 3 |
| Statistics | 4 | 11 |
| Architecture | 3 | 16 |

*3.2. Variables*

With input from the SSHPA database, the study contains the following variables:

- Author: the person who published the paper recorded into the SSHPA system. Author information includes general information (SSHPA ID, full name, year of birth, sex, nationality, Scopus Author ID), contact information, affiliation, and articles. In this article, sometimes the term 'Researcher' is used with the same meaning as 'Author'.

   ○ Lead author: The first author of a paper.
   ○ "New" author: Those who are recorded to have a publication for the first time in the SSHPA database.

- Publication: includes journal articles, books, book chapters, and conference proceedings in the NAFOSTED list mentioned above. During the data entry process, each publication is assigned to a certain field. Interdisciplinary publications can be assigned to more than one field.

   ○ Vietnamese authored publications: publications whose authors are all Vietnamese.
   ○ Internationally collaborated publications: publications that have a least one author whose nationality is not Vietnamese.

- Affiliation: the author's workplace, as stated in the publication.
- JIF: The impact factor level of the journal in which the articles were published. This varies from year to year.

In this paper, we employ descriptive analysis to examine the internal capabilities of Vietnam's SSH research (last updated 30 April 2020). The data include:

- The number of Vietnamese and foreign lead authors,
- The number of Vietnamese authored publications and internationally collaborated publications,
- The collaboration network of domestic institutes,
- The number of affiliations and articles in domestic research collaboration,
- The number of Vietnamese researchers by gender, including the number of all authors, the number of "new" authors, the number of lead authors, and the number of solo authors.

## 4. Results

### 4.1. The Scientific Output of Authors in Vietnam's SSH in Terms of Lead Authors and Gender Differences

In the 2008–2019 period (data as of 30 April 2020), the SSHPA system has recorded a total of 3160 publications written by 4021 authors, in which 2040 are Vietnamese authors and 1981 are foreign authors. Among 3160 publications, 2419 publications were led by Vietnamese authors (77.26%) and 712 publications were led by foreign authors (22.74%).

As can be seen from Figure 1, the number of Vietnamese lead authors increased steadily during the period from 2008–2019; the average annual growth rate of Vietnamese lead authors was approximately 22.33%, while that of foreign authors was 10.60%. In 2008, the ratio between the number of Vietnamese lead authors and foreign lead authors was 2.07; the ratio became 6.69 in 2019, an increase of over three-fold after eleven years.

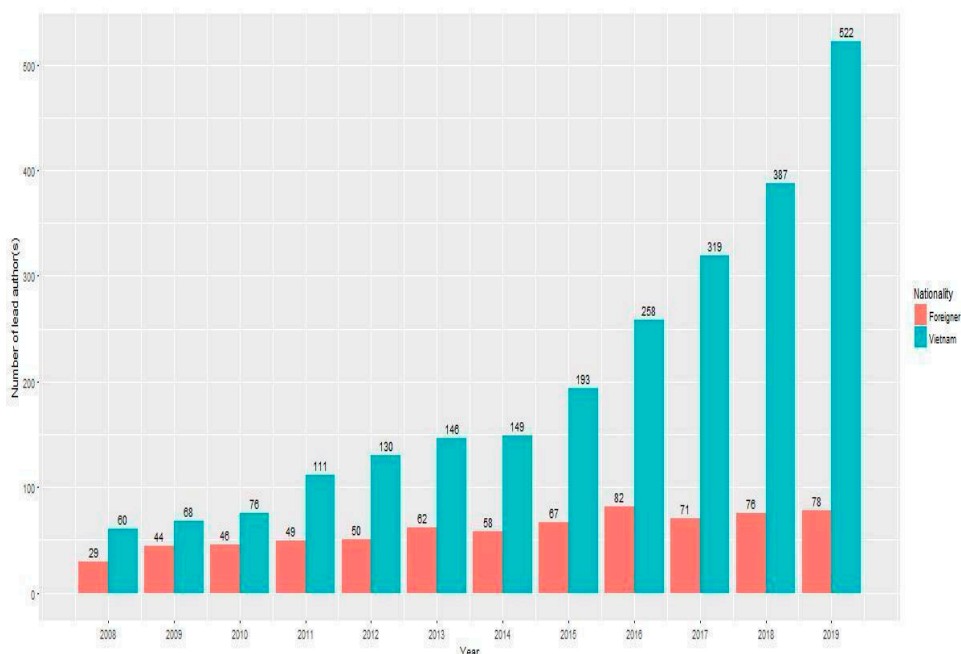

**Figure 1.** The number of Vietnamese and foreign lead authors in SSH research during 2008–2019 (last updated 30 April 2020).

In addition, within the 2008–2019 period, the number of male lead authors is consistently higher than that of women (see Figure 2). The level of difference has not changed much over the years. The only exception is that in 2018, the number of female and male lead authors was almost equal.

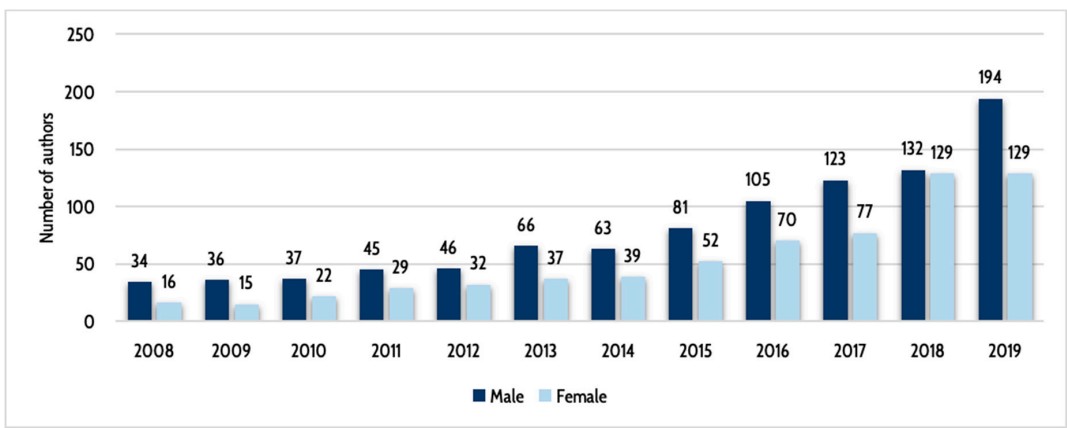

**Figure 2.** Number of Vietnamese lead authors by gender from 2008–2019 (last updated 30 April 2020).

Table 2 presents the annual proportion of the number of authors by gender during the period between 2008 and 2019. The dominance of male authors in SSH research is relatively clear; on average, male authors account for approximately two-thirds of the total author number during the period.

**Table 2.** Number of authors participating in Social Sciences and Humanities research by gender during the period from 2008–2019 (last updated 30 April 2020).

| Year | All Authors | Female Authors | | Male Authors | |
|---|---|---|---|---|---|
| | | No. of Authors | Percentage | No. of Authors | Percentage |
| 2008 | 138 | 53 | 38.41% | 85 | 61.59% |
| 2009 | 178 | 58 | 32.58% | 120 | 67.42% |
| 2010 | 182 | 59 | 32.42% | 123 | 67.58% |
| 2011 | 234 | 86 | 36.75% | 148 | 63.25% |
| 2012 | 281 | 102 | 36.30% | 179 | 63.70% |
| 2013 | 325 | 122 | 37.54% | 203 | 62.46% |
| 2014 | 355 | 131 | 36.90% | 224 | 63.10% |
| 2015 | 426 | 163 | 38.26% | 263 | 61.74% |
| 2016 | 554 | 215 | 38.81% | 339 | 61.19% |
| 2017 | 906 | 344 | 37.97% | 562 | 62.03% |
| 2018 | 1081 | 462 | 42.74% | 619 | 57.26% |
| 2019 | 1986 | 773 | 38.92% | 1213 | 61.08% |
| Average | | | 37.30% | | 62.70% |

In terms of "new" authors, from 2008 to 2017, there has been a consistently higher number of new male authors compared to their female counterparts. However, the average annual growth rate of new female researchers (21.41%) was higher than that of male researchers (16.78%). Interestingly, in 2018, women scientists did surpass their male colleagues by 12 "new" authors (Figure 3). With a similar number of female lead authors compared to male ones in the same year, 2018 appears to be a successful year for female researchers in Vietnamese SSH. In the following year, 2019, the differences in the number of "new" authors of both genders were quite small, hinting that female scientists might gradually improve their scientific productivity in years to come.

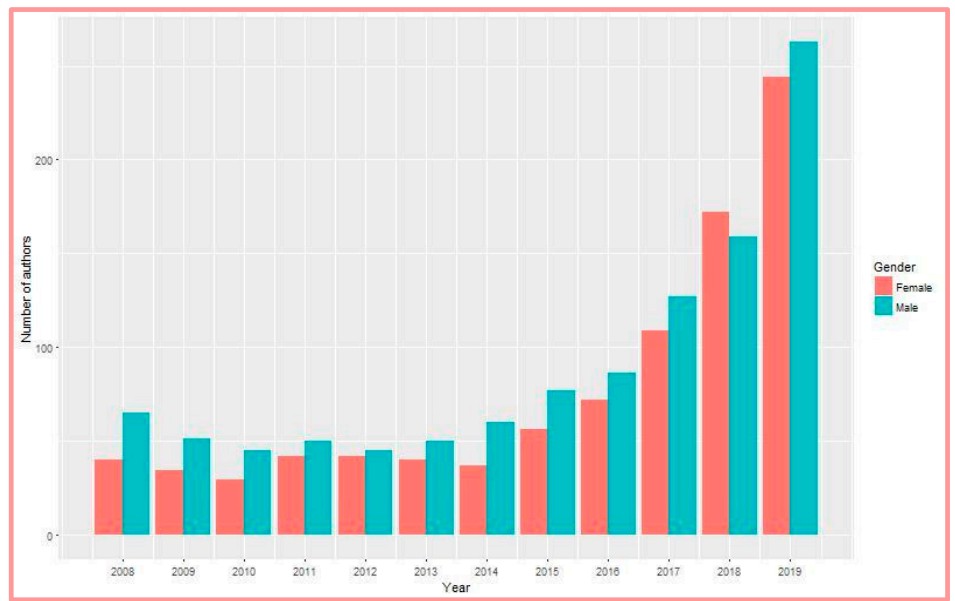

**Figure 3.** Number of "new" Vietnamese authors participating in SSH research by gender in the period from 2008–2019 (Last updated 30 April 2020).

During the period investigated, the number of solo authors has an overall increasing trend despite considerable fluctuations. 2010 is the year with the lowest number of solo authors (15 authors), while 2018 is the year with the highest number (81 authors) (Figure 4). This is also the only year that the number of female solo authors surpassed the number of male ones (44 female authors compared with 37 male authors). In terms of the results for lead and "new" authors, this consistently shows a positive result for the independent research capabilities of female authors.

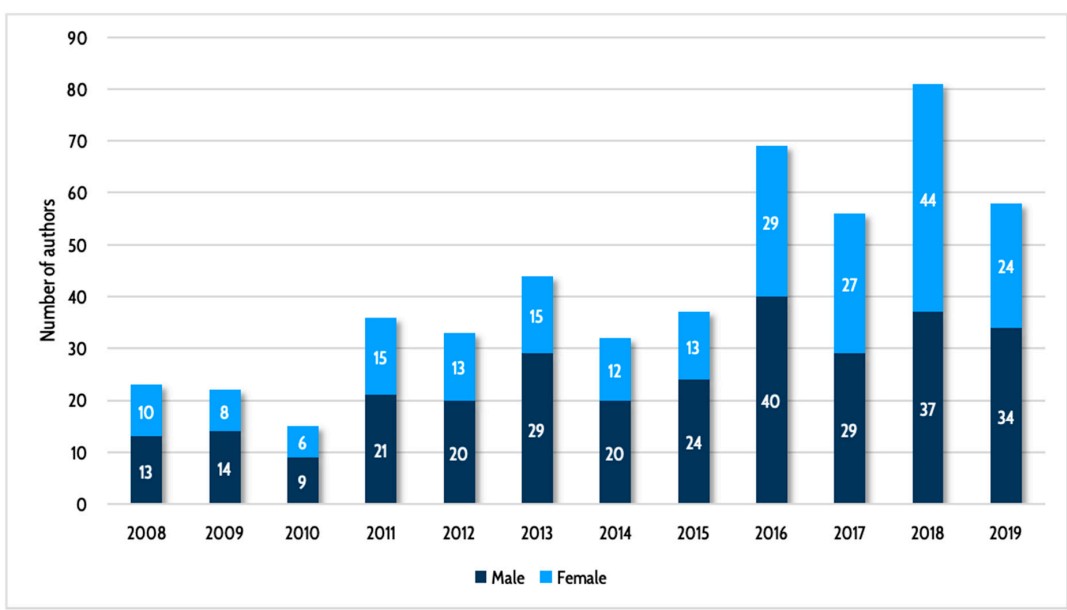

**Figure 4.** Number of solo authors by gender from 2008 to 2019 (last updated 30 April 2020).

*4.2. Research Collaboration Patterns of Authors in Vietnamese SSH*

Collaboration patterns are divided into two types: international collaboration and domestic collaboration. In terms of international collaboration, from 2008 to 2019, Vietnam cooperated in research with 77 countries around the world. The top 20 countries with the most cooperated publications are illustrated in Table 3. From the results, it can be seen that most of these top partners in terms

of research publications are developed nations with advanced research backgrounds, for example, Australia, the United States, or the United Kingdom. Notably, Australia is the most frequent partner to Vietnamese SSH, producing 855 publications during the period from 2008–2019, nearly 50% higher than the second partner, the United States. Given the dominance of developed countries' collaborating role, Thailand is an exception in the list, with nearly 80 publications co-authored with Vietnamese researchers. Finally, China, despite geographical proximity and long intertwined history, shares a limited number of publications with Vietnam's SSH disciplines.

**Table 3.** Top 20 countries in Vietnam research collaboration network during the period from 2008–2019 (last updated 30 April 2020).

| No | Country | Publication | No | Country | Publication |
|----|---------|-------------|----|---------|-------------|
| 1 | Australia | 855 | 11 | Thailand | 78 |
| 2 | United States | 578 | 12 | South Africa | 76 |
| 3 | Singapore | 208 | 13 | Netherlands | 67 |
| 4 | United Kingdom | 170 | 14 | Sweden | 63 |
| 5 | France | 151 | 15 | South Korea | 49 |
| 6 | Japan | 141 | 16 | Taiwan | 42 |
| 7 | Germany | 106 | 17 | China | 41 |
| 8 | Belgium | 98 | 18 | Denmark | 37 |
| 9 | New Zealand | 97 | 19 | Norway | 35 |
| 10 | Canada | 82 | 20 | Switzerland | 31 |

From 2008 to 2019, there were 1369 publications written by Vietnamese authors only and 1810 internationally collaborated publications. It is notable that within the three years from 2017 to 2019, the percentage of papers authored exclusively by Vietnamese people rose by a significant amount and even surpassed the percentage of internationally collaborated papers in 2019 for the first time, with 55.83% and 44.17%, respectively (see Table 4). This shows an improvement in the internal capability of Vietnamese SSH.

**Table 4.** Articles whose authors are all Vietnamese vs. articles that have non-Vietnamese authors during the period from 2008–2019 (last updated 30 April 2020).

| Year | Vietnamese Authored Publications | | Internationally Collaborated Publications | |
|------|----------------|------------|----------------|------------|
| | **No. of Articles** | **Percentage** | **No. of Articles** | **Percentage** |
| 2008 | 33 | 36.67% | 57 | 63.33% |
| 2009 | 35 | 30.17% | 81 | 69.83% |
| 2010 | 33 | 27.05% | 89 | 72.95% |
| 2011 | 67 | 41.88% | 93 | 58.13% |
| 2012 | 73 | 39.89% | 110 | 60.11% |
| 2013 | 81 | 38.57% | 129 | 61.43% |
| 2014 | 73 | 34.76% | 137 | 65.24% |
| 2015 | 92 | 35.38% | 168 | 64.62% |
| 2016 | 157 | 45.24% | 190 | 54.76% |
| 2017 | 190 | 46.34% | 220 | 53.66% |
| 2018 | 217 | 44.83% | 267 | 55.17% |
| 2019 | 378 | 55.83% | 299 | 44.17% |

In particular, Figure 5 illustrates that the discipline with the largest number of authors is Economics, which is also balanced in terms of the types of collaboration (581 publications collaborated domestically and 596 collaborated internationally). Health care is the field with the most significant difference (594 internationally collaborated papers and only 138 domestically collaborated papers). Out of 30 disciplines, there are 17 fields in which Vietnamese authors mainly works with domestic authors, most typically Demography, Economics, Asian Studies or Law.

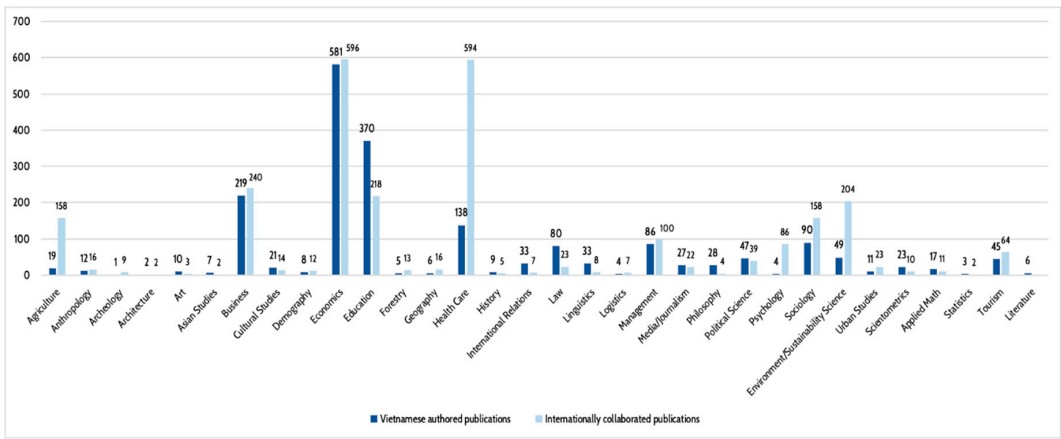

**Figure 5.** The number of publications domestically collaborated and internationally collaborated by discipline during the period from 2008–2019 (last updated 30 April 2020).

The period from 2016 to 2019 showed remarkable growth in the proportion of Vietnamese authored publications, which could be explained by the substantial expansion of the domestic collaboration network. Figure 6 illustrates in detail this domestic collaboration. Lines demonstrate collaboration times among provinces. In 2008, domestic collaborations only existed between research institutes in Hanoi, Ho Chi Minh City, and Long An province, and the number of collaborative publications was limited at 14 papers. In contrast, the collaboration network in 2019 has become much wider and denser, with a considerably higher number of collaborations between the three most developed cities (Ha Noi, Ho Chi Minh City, and Da Nang city) as well as other regions across Vietnam, from the North to the South. The collaboration between institutes in more and less developed areas also exists, such as collaborations between Ha Noi, Lam Dong, and Son La provinces.

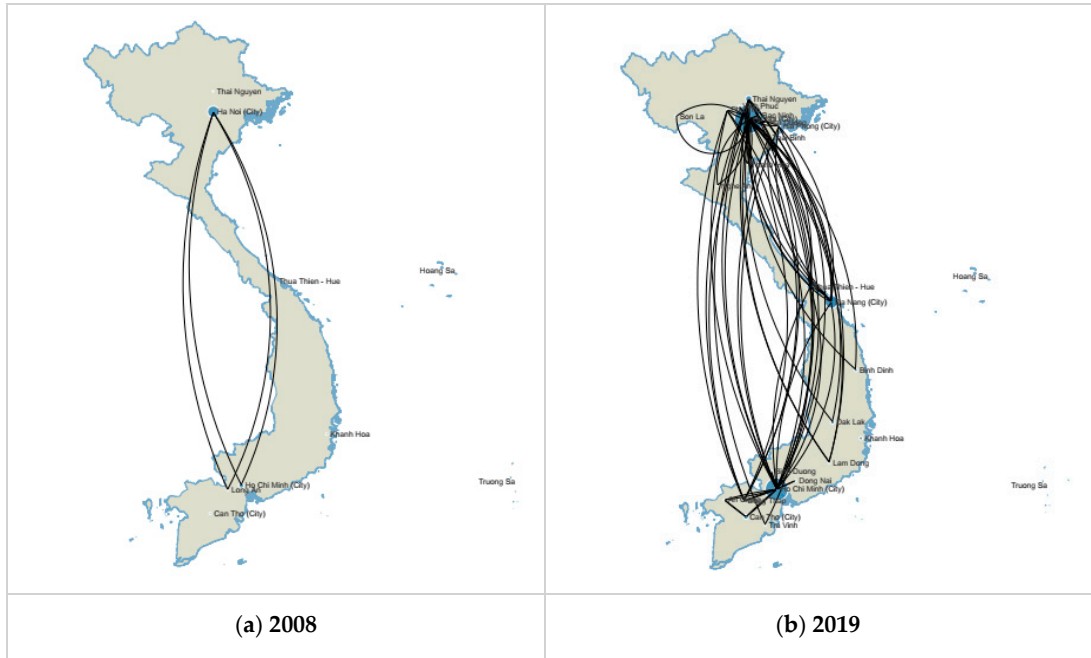

**Figure 6.** A map of Vietnamese domestic collaboration in (**a**) 2008 and (**b**) 2019 (last updated 30 April 2020).

To investigate the diversity and coverage of domestic collaboration, the number of affiliations collaborating with each other in Vietnam and their publications from 2008 to 2019 are illustrated in Figure 7. In general, the number of affiliations and articles has risen dramatically within the 11-year

period. The rapid and continuous growth started in 2016, and particularly in 2019, the numbers of affiliations and articles were almost double the figures of the previous year.

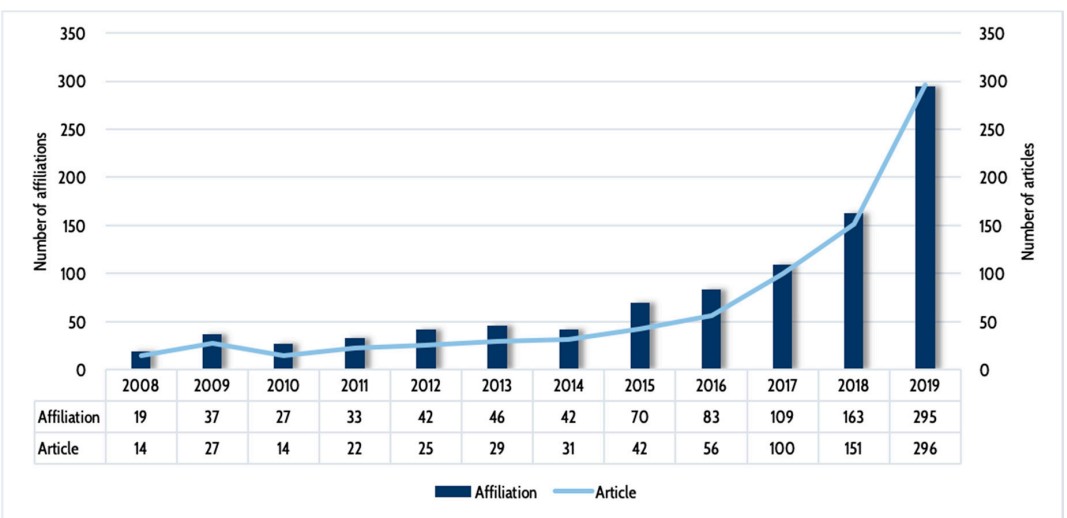

**Figure 7.** Numbers of affiliations and articles in domestic collaboration during the period from 2008–2019 (Last updated 30 April 2020).

## 5. Discussion

Through the data retrieved from the SSHPA database from 2008 to 2019, we examined the self-sustaining capacity of SSH research in Vietnam. Several main findings were observed. First, the internal research capacity in SSH fields in Vietnam has been increasing, as evidenced by a growing number of domestic lead authors and an expanding domestic research collaboration network. Second, even though the productivity and work quality of Vietnamese female researchers was undeniable, their participation in SSH research remained limited. Third, during the period, internationally collaborated research projects still dominated the scientific output of Vietnam's SSH; however, recent years have witnessed a strong rise in domestic authors and domestic networks. In 2019, the number of publications with exclusively Vietnamese authors even exceeded that figure of internationally collaborated ones.

### 5.1. Lead Authors, Solo Authors and Female Authors in Vietnam's SSH

Domestic SSH researchers have been asserting their capacity to grow in terms of output and adopting the lead author role. Of all 3131 publications, 77.26% were led by Vietnamese authors. Moreover, the ratio of Vietnamese lead authors to foreign lead authors has risen by more than three-fold, from 2.06 in 2008 to 6.69 in 2019. The number of solo authors also has an increasing trend. These results highlight the growing independence of Vietnamese SSH researchers in collaborations with foreign authors, especially during the period from 2017–2019, with a drop in the number of foreign lead authors compared to 2016. The high number of foreign authors in the 2008–2016 period might be a result of movement to study overseas for a long period [34]. As such, those who have studied overseas might publish their first papers with their advisors and colleagues oversea. They have then recently started to move back to Vietnam to train a new generation of researchers. As the leading researchers are trained in Western institutions, the promotion of international scientific publications in ISI/Scopus-indexed journals is inevitable. The proactive attitude of the Vietnamese government in the Circular 08/2017/TT-BGDDT, which outlines regulations stating that doctoral students need to publish in ISI/ Scopus-indexed journals, might have factored in the promising development of the Vietnamese academic community [35–37]. Moreover, Vietnamese universities are commonly making changes to upgrade their quality and engage with international ranking boards such as Times Higher

Education or QS. Reliance on the criteria of these ranking boards also contributes to the focus on scientific publications.

The current study also found that the participation rate of women in SSH research in Vietnam was modest; female researchers only accounted for 38.74% of all authors during the period from 2008–2019. Given the fact that the quality of research carried out by female authors was not dissimilar to that of male authors, an increase of women in science would greatly contribute to the enhancement of the research capacity and quality of Vietnam's SSH research [36]. Therefore, we recommend policymakers to consider financial allocations as well as special training programs that would alleviate the barriers to practicing science experienced by female academics and encourage their activity and constant participation in research activities. Currently, the adoption of new standards regarding sex and gender issues in science, such as the SAGER guidelines [38], would raise awareness of this issue among researchers. Moreover, promotion of women in leadership roles would also provide more opportunities to create equality in funding.

*5.2. Research Collaboration Patterns of Authors in Vietnam's SSH*

The collaboration patterns of SSH researchers are changing. We discovered a shift from heavily international networks to more localized networks in the last 4 years. After 2015, the percentage of exclusively Vietnamese authored publications grew rapidly and even surpassed the percentage of internationally collaborated publications in 2019, with 53,68%. This might have resulted, in part, from endeavors of the Vietnamese government (Resolution No.77/NQ-CP, 2014) to raise the research productivity of lecturers and faculties in public universities after granting them more autonomy but cutting down financial subsidies [39,40]. Although the decentralization reform in higher education in Vietnam has proven to be rather fruitful so far, one ought not to neglect practical problems and adverse impacts arising from conflict between extensive policies, organizational structures originating from and adapted from Western academic institutions, and Vietnamese cultural peculiarities [40–44].

We also found the emergence of collaboration between institutes in developed areas and less developed areas of Vietnam. As collaboration is an effective way to disseminate knowledge and skills via spillover, this is a step towards the goal of standardizing education, which is an aim of the centralized government of Vietnam. Nevertheless, the number of such collaborations is still limited. In certain areas in the country, such as the southernmost, the northernmost, and the central coasts, institutions have yet to forge a collaboration network with those in developed regions. Thus, future policymakers should pay more attention to incentivizing collaboration between developed areas and less developed areas to promote self-sustenance among all institutes nationwide. For instance, special funding mechanisms for less developed areas are crucial in promoting science in these areas. Moreover, research institutions from developed areas should also collaborate with those from less developed areas through training programs, field work, or even opening a new campus.

Promoting higher numbers of publications by incentivizing researchers to achieve some kind of "key performance indicators" might also have several consequences. It should be noted that in the context of an emerging economy with limited budget for science as a whole, financial resources should be allocated not only as a reward for high performers but also to develop training programs for early-career researchers, which is also a way to sustain high-quality human resource. The promotion of scientific productivity, which leads to an intense level of international collaboration, may also result in some dangers for SSH disciplines because research in these fields is usually rooted in local contexts. Therefore, cooperating with foreign authors, or even being led by these colleagues, may lead to a higher chance of incorporating modern theories and frameworks, most of which are from the Western world. This means that the diversity and originality of the research may be compromised. Finally, in order to achieve the funding, some researchers might choose shortcuts, which potentially lead to severe consequences such as retraction [45].

To conclude, after a period of heavy dependence on foreign authors and international collaborations, the field of SSH in Vietnam has experienced remarkable achievements in raising its domestic research

capacity, along with not-so-remarkable progress towards gender equality in science. In order to further promote the capacity of SSH research and the Vietnamese academic community, as well as to enhance public trust [30,46], future policy-makers need to continue promoting domestic research and collaboration among Vietnamese researchers and institutes nationwide and empowering women in SSH research.

## 6. Limitations

Despite the contribution to the understanding of the internal capability of Vietnamese SSH, we would like to acknowledge several limitations of this article. Firstly, we approach the subject of internal research capacity from a sheer quantitative approach, focusing on ISI/Scopus-indexed journals. The journals, books or conference proceedings that are indexed in the ISI or Scopus database are potentially biased toward English language and Western-oriented publications. Even though the usage of these databases is common practice [4,5], the exclusion of scientific publications in other languages and journals might hinder thorough evaluation of Vietnamese SSH researchers. Moreover, social sciences and humanities are qualitative and hermeneutic disciplines. Therefore, the usage of a quantitative approach and the act of counting publications only present an initial effort to reach a perspective on Vietnamese SSH. In future research studies, different approaches should be used to discuss the issue more critically.

**Supplementary Materials:** Supplementary Materials can be found at http://www.mdpi.com/2304-6775/8/2/32/s1.

**Author Contributions:** Conceptualization, M.-T.H. and Q.-H.V.; Data curation, M.-T.H.; Funding acquisition, Q.-H.V.; Investigation, T.-T.V., T.-H.P.; Methodology, Q.-H.V.; Project administration, M.-T.H.; Resources, Q.-H.V.; Supervision, Q.-H.V.; Validation, M.-T.H., T.-H.P., T.-T.V.; Visualization, A.-P.L. and T.-N.N.; Writing—original draft, A.-P.L. and T.-N.N.; Writing—review & editing, M.-T.H., T.-H.P., T.-T.V. All authors have read and agreed to the published version of the manuscript.

**Funding:** This research is funded by the Vietnam National Foundation for Science and Technology Development (NAFOSTED) under the National Research Grant No. 502.01-2018.19.

**Conflicts of Interest:** The authors declare no conflict of interest.

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
