# Peer review of "The Internal Capability of Vietnam Social Sciences and Humanities: A Perspective from the 2008–2019 Dataset"

_publications, doi:10.3390/publications8020032_

Round 1

Reviewer 1 Report

The article is original and of scientific interest. The research method used is consistent and appropriate. The results are consistent with the intended objectives. The bibliography used is appropriately focused

Author Response

Letter of detailed responses to Reviewer 1 

Dear Sir/Madam, 

Thank you very much for your time and efforts in reviewing our manuscript. We highly appreciate your hard work and kind comments:

The article is original and of scientific interest. The research method used is consistent and appropriate. The results are consistent with the intended objectives. The bibliography used is appropriately focused

Please accept our sincere thanks for your great contributions to the improvement of financial studies and the overall advancement of sciences in the world. 

Shall you have further comments, we look forward to hearing from you.  

Best regards, 

Reviewer 2 Report

The main problem of this article is the lack of contextualization of the theme, which is so vital and distinctive in the vocation of the social sciences and humanities. The social and cultural reality of Vetnam is not sufficiently addressed, so full understanding of the object of study become difficult.
The study aims “to examine the domestic research capability in Social Science & Humanities (SSH) disciplines through the number of Vietnamese lead authors, domestic 21 collaboration network, and the proportion of women in science”
In view of this purpose, it is strange that there are no answers or developments to expected topics and themes:

What particularities distinguish “the fields of SSH in Vietnam”? Are they a homogeneous group? Which countries are included in the "international networks"? What is the real importance of determining trends in terms of authorship of  3,138 publications (scientific articles) during the period 2008-2019 without knowing the topics and the fields they address?

How has the presence of women evolved in various stages of the production of scientific knowledge: from teaching to research practice, including the performance of management functions in schools, professional associations and other institutions? The authors argue that policy makers should empower female researchers but the obstacles they face (in training, education, international mobility, political participation, salary, family and professional life combination) deserve little development.
This results in the almost exclusively descriptive tone of the article and also in the lack of distinction between the discussion and the conclusion part.
There are various relationships that could be established between different trends and generate a more reflective approach.
The article would gain in reflecting the bibliography in a more in-depth way.

Author Response

Letter of detailed responses to Reviewer 2 

Dear Sir/Madam, 

Thank you very much for your time and efforts in reviewing our manuscript. Your detailed comments have helped us a lot to improve the quality of our paper.  

We have addressed your points in our revised version. Please note that in the revised paper, the parts highlighted in yellow are for correction on the old text and the parts highlighted in green are written anew. Below are our answers to your comments (in italic). Also, the line numbers in the text refer to the revised paper. 

The main problem of this article is the lack of contextualization of the theme, which is so vital and distinctive in the vocation of the social sciences and humanities. The social and cultural reality of Vietnam is not sufficiently addressed, so full understanding of the object of study become difficult.

Thank you for your detailed comments, we have rewritten the introduction part to better set the context for the study as follows:

  1. Introduction

In the era of globalization, international collaboration is an inevitable trend in many areas, including science. The cooperation with researchers from other countries is shown to improve absolute scientific output [1], and this is of critical importance for developing nations to enhance the performance and capabilities of their research communities. For example, in South and East Asia, international collaboration has a very substantial impact and it is this activity that raises their average citation impact above the global average [2]. However, the dependence on this international collaboration, to some extent, may also have negative impacts on the sustainability of a country’s scientific productivity and quality. Therefore, building internal capabilities of scientists in a national context is imperative especially for developing countries.

Vietnam, as an emerging economy, has witnessed the rapid development of science over the past decades [3]. The Global Research Report by Web of Science (2019) states that in South and East Asia, Vietnam has the fastest growth rate with its indexed publication volume more than five-fold since 2009 and shows every sign of continuing expansion [2]. Despite this seemingly positive information, international collaboration in fact accounted for approximately 77% of all the publications [4,5].  Noticeably, the majority of these international projects were led by foreign authors [4]. From another perspective, studies utilizing network analysis to analyze co-authorship patterns among Vietnamese researchers indicate some signals of unsustainability. These studies found that in any given scientific group, most co-authors are centered around well-connected and highly productive researchers [5] and there are some indicators of unsustainable co-authorship groups due to their big gaps in productivity and connection [6].

In the regional context, among East and Southeast Asian countries, Vietnam still has low research intensity primarily due to its heavy reliance on foreign co-authors and limited quantity of international publications in applied and multidisciplinary fields [7]. During the 1991 – 2010 period, Vietnam researchers’ productivity is only equivalent to 13.33% of Singapore’s and 29% of Thailand’s [8,9].

Among various disciplines, social sciences and humanities (SSH) appear to have a humble position compared with other fields in Vietnam and have been disparaged for their low productivity in both the total number and the adjusted number in comparison with natural science and technology disciplines [3,10].  By definition, SSH is the study of human behavior and interaction in social, cultural, environmental, economic, and political contexts [11]. Currently, SSH is not only implemented in the context of each separate discipline; instead, contemporary SSH projects are inherently interdisciplinary, taking place in different scientific and organizational environments [12]. Compared to natural sciences, SSH is more local-oriented [13]. Therefore, in the context of Vietnam’s fast-growing economy, SSH plays an important role in investigating the sociocultural as well as economic situations to provide reliable implications for policy makers as well as practitioners.

Given the fact that Vietnamese authors still need to cooperate with international partners in scientific projects while attempting to enhance the domestic scientific sustainability, this article aims at providing a comprehensive view about the internal capabilities of the country’s SSH disciplines in the 2008 - 2019 period. This also provides implications for policy making as well as social scientists to contribute to the advancement of science of the nation.

The study aims “to examine the domestic research capability in Social Science & Humanities (SSH) disciplines through the number of Vietnamese lead authors, domestic 21 collaboration network, and the proportion of women in science”

In view of this purpose, it is strange that there are no answers or developments to expected topics and themes:

What particularities distinguish “the fields of SSH in Vietnam”? Are they a homogeneous group? Which countries are included in the "international networks"? What is the real importance of determining trends in terms of authorship of  3,138 publications (scientific articles) during the period 2008-2019 without knowing the topics and the fields they address?

We have included the list of SSH fields in the 3.1. Materials part:

The SSHPA system collects paper from 31 fields in SSH research. Detailed information of these disciplines was listed in Table 1.

Table 1: Disciplines in Social Sciences and Humanities in Vietnam (2008 – 2019)

Field

Number of Articles

Number of Authors

Economics

941

1350

Education

484

534

Health Care

451

967

Business

362

580

Sociology

213

435

Environment/Sustainability Science

202

538

Management

154

269

Agriculture

143

326

Law

95

79

Tourism

90

121

Political Science

84

115

Psychology

63

189

Media/Journalism

40

53

Linguistics

39

33

International Relations

38

31

Philosophy

30

74

Cultural Studies

30

66

Urban Studies

27

55

Anthropology

24

44

Applied Math

23

39

Scientometrics

18

43

Geography

18

57

Forestry

15

33

Demography

15

32

History

13

21

Art

11

24

Logistics

10

18

Asian Studies

7

16

Archeology

7

36

Literature

6

3

Statistics

4

11

Architecture

3

16

We also added the list of top 20 countries that have collaboration in terms of research with Vietnam as follows:

Table 2: Top 20 countries in Vietnam research collaboration network during 2008 - 2019 (Last updated 30 April 2020)

No

Country

Publication

No

Country

Publication

1

Australia

855

11

Thailand

78

2

United States

578

12

South Africa

76

3

Singapore

208

13

Netherlands

67

4

United Kingdom

170

14

Sweden

63

5

France

151

15

South Korea

49

6

Japan

141

16

Taiwan

42

7

Germany

106

17

China

41

8

Belgium

98

18

Denmark

37

9

New Zealand

97

19

Norway

35

10

Canada

82

20

Switzerland

31

How has the presence of women evolved in various stages of the production of scientific knowledge: from teaching to research practice, including the performance of management functions in schools, professional associations and other institutions? The authors argue that policy makers should empower female researchers but the obstacles they face (in training, education, international mobility, political participation, salary, family and professional life combination) deserve more development.

We have added several points In the literature review part to address your comment as follows:

In Vietnam, the long-standing Confucius society has long considered academic achievements more or less men’s role, while perceiving women’s key responsibilities mostly in terms of housework. Until 2010, women made up only 11% of PhD and masters, 1% of professors and 4% of  associate professors [22]. Such low prevalence of female researchers has been attributed to gender discrimination in terms of salary, training, promotion, research funding, and committee member appointment [23]. Despite gender equality in science being a crucial goal, few studies have examined this matter in SSH research in Vietnam. Among those few, Vuong et al. [17] found that there is virtually no difference between male and female contribution in Vietnamese SSH research. Therefore, encouraging more women to participate in scientific research would greatly benefit the scientific performance in Vietnam through improving research productivity, flexibility, accuracy, and innovation [24,25]. In other words, gender diversity is crucial in expanding the internal capabilities of doing science in Vietnam.

This results in the almost exclusively descriptive tone of the article and also in the lack of distinction between the discussion and the conclusion part.

We have restructured the Discussion part to be consistent with contents of the Literature Review and Results parts, and use the Conclusion part as a summary of key points of the paper as follows:

  1. Discussion

Through the data retrieved from the SSHPA database from 2008 to 2019, we examined the self-sustaining capacity of SSH research in Vietnam. Several main findings were observed. First, the internal research capacity in SSH fields in Vietnam had been building up, as evidenced by a growing number of domestic lead authors and an expanding domestic research collaboration network. Second, even though the productivity and work quality of Vietnamese female researchers was undeniable, their participation SSH research remained limited. During the period, internationally collaborated research projects still dominate the scientific output of Vietnam’s SSH; however, the recent years have witnessed a strong rise of domestic authors and domestic network.  In 2019, the number of publications with exclusively Vietnamese authors even exceeded that figure of international collaborated ones.

5.1. Lead authors, solo authors and female authors in Vietnam’s SSH

Domestic SSH researchers have been asserting their capacity to grow in terms of output and playing the lead author role. Out of 3,131 publications, 77.26% was led by Vietnamese author. Moreover, the ratio of Vietnamese lead authors to foreign lead authors has risen by more than three-fold, from 2.06 in 2008 to 6.69 in 2019. The number of solo authors also has an increasing trend. These results highlight the growing independence of Vietnamese SSH researchers from collaborations with foreign authors, especially in 2017-2019 with a drop in the number of foreign lead authors compared to 2016. The high number of foreign authors in the 2008-2016 period might be an impact of the focus on sending students to study oversea for a long period [Cite policy]. Those who have studied oversea might published their first papers with their advisors and colleagues oversea. Then, only until recently that they have started to move back to Vietnam to train a new generation of researchers. As the leading researchers are trained in the Western institutions, the promotion of international scientific publications in ISI/Scopus-indexed journals are inevitable. The proactive attitude of Vietnamese government through the Circular 08/2017/TT-BGDDT, which outlines the regulations on doctoral enrolment and training might have factored into the promising development of the Vietnamese academic community [34-36]. Moreover, as the Vietnamese universities are making changes to upgrade the quality, the usage of international ranking boards such as Times Higher Education or QS is common. The reliance on the criteria of these ranking boards also contribute to the focus on scientific publications.

The current study found that the participation rate of women in SSH research in Vietnam was modest; female researchers only accounted for 38.74% of all authors during the period 2008-2019. Given the fact that the quality of research done by female authors was not dissimilar to that of male authors, an increase of women in science would greatly contribute to the enhancement of research capacity and quality of Vietnam’s SSH research [35]. Therefore, we recommend policymakers to consider financial allocations, special training programs in such a way that would alleviate barriers in doing science for female academics and encourage their active, constant participation in research activities. Currently, the adoption of new standards regarding sex and gender issues in science such as SAGER guidelines [37] would raise the awareness of researchers about the issues. Moreover, promotion of women in leadership would also provide more opportunities to create equal in funding.

5.2. Research collaboration patterns of authors in Vietnam’s SSH

The collaboration patterns of SSH researchers are changing. We discovered a shift from heavily international networks to more localized networks in the last 4 years. After 2015, the percentage of exclusively Vietnamese-authored publications grew rapidly and even surpassed the percentage of internationally collaborated publications in 2019 with 53,68%. This might have resulted, in part, from endeavors of Vietnamese government (Resolution No.77/NQ-CP, 2014) in raising research productivity of lecturers and faculties in public universities after granting them more autonomy but cutting down financial subsidies [38,39]. Although the decentralization reform in higher education in Vietnam have proven to be rather fruitful so far, one ought not to neglect practical problems and adverse impacts due to the conflict between extensive policies, organizational structures originating from and adapted from the Western academic institution and Vietnamese cultural peculiarities [39-43].

We also found that the emergence of collaboration between institutes in developed areas and less developed areas in Vietnam. As collaboration is an effective way to disseminate knowledge and skills via spillover, this is a step toward the goal of standardizing education as aimed for by the centralized government of Vietnam. Nevertheless, the number of such collaborations is still limited. In certain areas in the country, such as the southernmost, the northernmost, and the central coasts, institutions have yet to forge a collaboration network with any institution in developed regions. Thus, future policymakers should pay more attention to incentivize collaboration between developed areas and less developed areas to promote self-sustenance among all institutes nationwide. For instance, special funding mechanisms towards less developed areas are crucial in promoting science in these areas. Moreover, research institutions from developed areas should also collaborate with the counterparts from less developed areas through training programs, field works, or even opening new campus.

From another aspect, the act of promoting higher number of publications by incentivizing researchers to achieve some kinds of “key performance indicators” might also have several consequences. It should be noted that within an emerging economy context with limited budget for science as a whole, the financial resource should be allocated not only for rewarding high-performers but also to develop training programs for early-career researchers, which is also a way to sustain the high-quality human resource. The promotion of scientific productivity, which leads to an intense level of international collaboration, may also result in some dangers for SSH disciplines, as the research in these fields are usually rooted in the local contexts. Therefore, cooperating with foreign authors, or even being led by these colleagues may lead to a higher chance of incorporating modern theories and frameworks, most of which from the Western world. This means that the diversity and originality of the research may be compromised.

To conclude, after a period of heavy dependence on foreign authors and international collaborations, the fields of SSH in Vietnam have had remarkable achievements in raising domestic research capacity, along with not-so-remarkable progress towards gender equality in science. In order to further promote SSH research capacity and the academic community, as well as to enhance public trust [30,44], future policy-makers need to continue promoting domestic research and collaboration among Vietnamese researchers and institutes nationwide and empowering women in SSH research.

There are various relationships that could be established between different trends and generate a more reflective approach.

The article would gain in reflecting the bibliography in a more in-depth way.

This comment is of significant value to us, based on it, we extracted additional data and compared results in a number of ways to have a more in-depth perspective. For example, we present results about lead authors as well as solo and new authors to highlight the internal capabilities of Vietnam’s SSH researchers. In terms of investigating gender disparity, the data expanded to include not only female and male researchers’ output, but also consider females’ role as lead authors, solo authors or new authors. We also added the list of top foreign country who have engaged intensively in research collaboration with Vietnam. In terms of domestic collaboration,  we added Figure 4 which presents the number of affiliation and article in domestic collaboration during 2008-2019.

Please see all the changes in the Results part as follows:

  1. Results

4.1. Scientific output of authors in Vietnam SSH in terms of lead authors and gender differences

In the 2008-2019 period (data as of 30 April, 2020), SSHPA system has recorded a total of 3,160 publications written by 4,021 authors, in which 2,040 are Vietnamese authors and 1,981 are foreign authors. Among 3,160 publications, 2,419 publications were led by Vietnamese authors (77.26%) and 712 publications were led by foreign authors (22.74%).

As can be seen from Figure 1, the number of Vietnamese lead authors increased steadily during the period 2008-2019; the average annual growth rate of Vietnamese lead authors was approximately 22.33%, while that of foreign authors was 10.60%. In 2008, the ratio between the number of Vietnamese lead authors and foreign lead authors was 2.07; the ratio became 6.69 in 2019, an increase of over three-fold in eleven years later

Figure 1: The number of Vietnamese and foreign lead authors in SSH research during 2008-2019 (Last updated 30 April, 2020)

In addition, within the 2008 - 2019 period, the number of male lead authors is consistently higher than their counterparts. The level of difference has not changed much over the years. The only exception is that in 2018, the number of female and male lead authors are almost equal.

Figure 6: Number of Vietnamese lead author by gender from 2008 to 2019 (Last updated 30 April, 2020)

Table 2 presents the annual proportion of the number of authors by gender during the period between 2008 and 2019. The dominance of male authors in SSH research is relatively clear; on average, male authors account for approximately two thirds of the total author number in the period (see Table 4).

Table 4: Number of authors participating in Social Sciences and Humanities research by gender during 2008-2019 (Last updated 30 April, 2020)

Year

All authors

Female authors

Male authors

No. of authors

Percentage

No. of authors

Percentage

2008

138

53

38.41%

85

61.59%

2009

178

58

32.58%

120

67.42%

2010

182

59

32.42%

123

67.58%

2011

234

86

36.75%

148

63.25%

2012

281

102

36.30%

179

63.70%

2013

325

122

37.54%

203

62.46%

2014

355

131

36.90%

224

63.10%

2015

426

163

38.26%

263

61.74%

2016

554

215

38.81%

339

61.19%

2017

906

344

37.97%

562

62.03%

2018

1081

462

42.74%

619

57.26%

2019

1986

773

38.92%

1213

61.08%

Average

37.30%

62.70%

In terms of new authors, from 2008 to 2017, there has been a consistently higher number of new male authors compared to their female counterparts. However, the average annual growth rate of new female researchers (21.41%) was higher than that of male researchers (16.78%). Interestingly, in 2018, women scientists did surpass their colleagues with the number by 12 new authors (Figure 5). With a similar number of female lead authors compared to male ones in the same year, 2018 appears to be a successful year for female researchers in Vietnam SSH. In the following year, 2019, the differences in the number of new authors of both genders is quite small, hinting that female scientists may gradually improve their scientific productivity in the recent years.

Figure 5: Number of “new” Vietnamese authors participating in SS&H research by gender in the period 2008 - 2019 (Last updated 30 April, 2020)

During the period investigated, the number of solo authors has an overall increasing trend despite considerable fluctuations. 2010 is the year with the lowest number of solo authors (15 authors) while 2018 is the year with the highest number (81 authors) (Figure 7). This is also the only year that the number of female solo authors surpassed the male ones (44 female authors compared to 37 male authors). Consistently with the results for lead and new authors, this shows a positive signal for the independent research capabilities of female authors.

Figure 7: Number of solo authors by gender from 2008 to 2019 (Last updated 30 April, 2020)

4.2. Research collaboration patterns of authors in Vietnam SSH

The collaboration patterns are divided into two types: international collaboration and domestic one. In terms of international collaboration, from 2008 to 2019, Vietnam cooperated in research with 77 countries around the world. The top 20 countries with the most cooperated publications are illustrated in Table 2. From the results, it can be seen that most of these top partners in terms of research publications are developed nations with an advanced research community, for example Australia, United States or United Kingdom. Notably, Australia is the largest partner with Vietnam SSH, producing 855 publications during the period 2008 – 2019, a level of nearly 50% higher compared to the second partner, United States. Given the dominance of developed countries’ collaborating role, Thailand is an exception in the list, with nearly 80 publications co-authored with Vietnamese researchers. Finally, China, despite geographical proximity and long intertwined history, shares a limited number of publications with Vietnam’s SSH disciplines.

Table 2: Top 20 countries in Vietnam research collaboration network during 2008 - 2019 (Last updated 30 April 2020)

No

Country

Publication

No

Country

Publication

1

Australia

855

11

Thailand

78

2

United States

578

12

South Africa

76

3

Singapore

208

13

Netherlands

67

4

United Kingdom

170

14

Sweden

63

5

France

151

15

South Korea

49

6

Japan

141

16

Taiwan

42

7

Germany

106

17

China

41

8

Belgium

98

18

Denmark

37

9

New Zealand

97

19

Norway

35

10

Canada

82

20

Switzerland

31

From 2008 to 2019, there were 1369 publications written by Vietnamese authors only and 1810 internationally collaborated publications. It is notable that within three years from 2017 to 2019, the percentage of papers authored exclusively by Vietnamese rose by a significant amount and even surpassed the percentage of international collaborated papers the first time in 2019 with 55.83% and 44.17% respectively (see Table 3). This shows an improvement in the internal capability of Vietnam SSH.

Table 3: Number of articles whose authors are all Vietnamese vs Articles that have non-Vietnamese authors during 2008-2019 (Last updated 30 April, 2020)

Year

Vietnamese authored publications

Internationally collaborated publications

No. of articles

Percentage

No. of articles

Percentage

2008

33

36.67%

57

63.33%

2009

35

30.17%

81

69.83%

2010

33

27.05%

89

72.95%

2011

67

41.88%

93

58.13%

2012

73

39.89%

110

60.11%

2013

81

38.57%

129

61.43%

2014

73

34.76%

137

65.24%

2015

92

35.38%

168

64.62%

2016

157

45.24%

190

54.76%

2017

190

46.34%

220

53.66%

2018

217

44.83%

267

55.17%

2019

378

55.83%

299

44.17%

In particular, the discipline with the largest number of authors is Economics, which also has a rather balance in the types of collaboration (581 authors collaborate domestically, and 596 authors publishing with international authors). Health care is the field with the most significant difference (594 internationally collaborated authors; and only 138 domestically collaborated authors). Out of 30 disciplines, there are 17 fields in which Vietnamese authors mainly works with domestic authors, typically Demography, Economics, Asian Studies, Law...

Figure 2: The number of Vietnamese authors in domestically collaborated publications and international collaborated publications by discipline during 2008-2019 (Last updated 30 April, 2020)

The period from 2016 to 2019 shows a remarkable growth in the proportion of Vietnamese authored publications, which could be explained by the substantial expansion of the domestic collaboration network. Figure 3 illustrates in detail this domestic collaboration. One line shows one collaborated publication between two affiliations in two provinces. Double lines demonstrate multiple collaboration times (at least twice). In 2008, domestic collaborations only existed between research institutes among Hanoi, Ho Chi Minh City, and Long An province, and the number of collaborative publications were limited at 14 papers. In contrast, the collaboration network in 2019 became much wider and denser, with a considerably higher number of collaborations between the three most developed cities (Ha Noi, Ho Chi Minh City, and Da Nang city) as well as other regions across Vietnam, from the North to the South. The collaboration between institutes in more and less developed areas provinces also exists, such as the collaborations among Ha Noi, Lam Dong and Son La provinces.

2008

2019

Figure 3: A map of Vietnam domestic collaboration a) 2008 and b) 2019 (Last updated 30 April, 2020)

To investigate the diversity and coverage of domestic collaboration, the number of affiliations collaborating with each other in Vietnam and their publications from 2008 to 2019 are illustrated in Figure 4. In general, the number of affiliations and articles have gone up dramatically in 11 years. The rapid and continuous growth started in 2016; specially; in 2019, both affiliations and articles almost double those figures of the previous year.

Figure 4: Number of affiliation and article in domestic collaboration during 2008-2019 (Last updated 30 April, 2020)

In closing, we highly appreciate the hard work and time that you have spent on this manuscript. Your comments have helped us a lot in improving the quality of our paper. We hope that the revised paper has met your requirements.  

Please accept our sincere thanks for your great contributions to the improvement of financial studies and the overall advancement of sciences in the world. 

Shall you have further comments, we look forward to hearing from you.  

Best regards, 

Reviewer 3 Report

This is an interesting study that looks at the development of authorship in Vietnam. It gives some perspectives on domestic collaboration and internationalisation, and relates temporal differences to policy issues. Moreover, it takes a strong perspective on the development and situation of female authors. The authors make clear what the source for their analysis is and what methods are applied.

However, the authors miss to clarify what the numbers alone should represent for readers cannot relate these in relative analyses. How is this a special case of Vietnam in relation to other ASEAN countries, is this a special issue of authorship development in relation to the development of researcher employment in Vietnam, and what are the particularities of SSH in respect to other disciplines? Such issues remain opaque, unfortunately, for the numbers are presented as such. Besides the data provided, the authors engage quite uncritically with it. It would make the submission much stronger if the authors track potential downsides on the sheer quantitative approach especially in regard to Western-indexed journals, the JIF, and the particularities of the analysed disciplines. I go into more detail about these issues in the following points. Some of these, especially the terminological corrections, seem necessary for publication. The latter points about potential inclusions of relative numbers and lines of discussion would strengthen the paper, but may not be necessary as such; they are more proposals to consider, which I hope are helpful.

  • What disciplines are subsumed under SSH. Did this change over the time the data represents?
  • 2: “only 740 (12.2%) of the 6,031 cited articles” > Could you relate this to the proportion of researchers in Vietnam? 12% does seem low intuitively. But maybe the articles are exceptional in themselves and so the low number is made up for with quality. Moreover, can you ascertain whether there are more than 12% researchers in SSH disciplines in Vietnam so that the authorship number indeed deviates in relation?
  • 2: NAFOSTED was established to “incentivize researchers” to do what? Is publishing more an incentive and measure of success?
  • Figure 2: This is an interesting representation. Yet, you do not provide an account for the numbers behind it as well as a legend clarifying what the double connection (double lines between sites of knowledge production) stands for. It would be good to have more information about this, especially what you refer to with “national collaboration network”. Are the researchers just collaborating more or are there new researchers and new institutions which initiates the increase by default?
  • The description of figures in table 2 is a bit unclear in that the terms “author” and “researcher” seem to be mixed up.
    • “among the total 6,401 researchers during the period 2008-2019” > This total denotates the total of authors, correct?
    • The description says “female researchers accounted for” while the table says “female authors”. I assume the latter is correct. If so, could you rectify this in the text?
    • What is the nominative here? I.e. what can the numbers tell us beyond mere authorship, is there are relative measure?
      • Is it possible to sort out repeated authorship for these numbers?
      • Again here, after clarifying on the terminology, it would be good to relate the two categories (institutionalised researchers and authors), if possible.
    • Table 3: Can you field-normalise these numbers? The JIF as such should not be used as a general metric of quality. If at all, it should only be used as a comparative metric within a field, looking at generalisable citation patterns. You objectify the quality of individual authors by means of generalised JIF numbers, which seems inappropriate.
    • Meaning of the numbers raised:
      • For instance in regard to gender, the reader cannot infer too much meaning since gender among authorship is something different than gender among researchers in Vietnam. As SSH are qualitative and hermeneutic disciplines, it may quite as well be the case that crude more of publications is not necessarily good, so inferring a critique based on authorship numbers alone may be short-sighted.
      • In their current single representation, the numbers are not quite relatable, neither normatively regarding native researcher growth, nor relatively regarding other clusters of disciplines. It wold be good, therefore, to include such comparisons in the study. If such comparisons are not available, you can hardly infer particularities of authorship in the SSH fields.
      • If there are only authorship numbers available and no numbers of researcher and institutional growth to relate these to, it would be good to at least provide comparisons to other disciplines.
      • On page 7, you discuss the scholarly success and “collaboration is an effective way to disseminate knowledge.” And yet, it is questionable what the role of authorship in these indexed journals is. Since the indices you mention are usually attributed a Western bias, do the journals also have such bias? If so, what does this mean for scholarship in Vietnam? What role has international authorship visibility in the national context of developing research policies. This may be an important discussion to be included since you base your analysis on authorship alone.
      • Furthermore, in the same line of discussion may be a perspective on analysing authorship as a placeholder for scholarly achievement. Since you are working on SSH scholarship, what could go wrong if policies incentivise sheer higher numbers of publications irrespective of correspondingly developing research programmes? Why is this particularly a danger for scholarship in SSH disciplines? What may be dangers of internationalising authorship particularly in humanities disciplines for they are usually rooted in regional scholarship?
    • There are some language issues, especially regarding word choice and sentence structure.

Many thanks for the paper and the request to have me review it!

Author Response

Dear reviewer 3,

Once again we would like to express our most sincere thanks for your contribution in helping us improve our submission. We have done major revisions following your very detailed input. The changes brought to the manuscript have been highlighted in yellow, whereas newly added passages were highlighted in green. In this letter, we will address each of the points you have made individually.

  • This is an interesting study that looks at the development of authorship in Vietnam. It gives some perspectives on domestic collaboration and internationalisation, and relates temporal differences to policy issues. Moreover, it takes a strong perspective on the development and situation of female authors. The authors make clear what the source for their analysis is and what methods are applied.

Thank you for your kind comments.

  • However, the authors miss to clarify what the numbers alone should represent for readers cannot relate these in relative analyses. How is this a special case of Vietnam in relation to other ASEAN countries, is this a special issue of authorship development in relation to the development of researcher employment in Vietnam, and what are the particularities of SSH in respect to other disciplines?

Thank you for your insightful suggestion. We have explained the case of Vietnam more thoroughly in the Introduction section:

Vietnam, as an emerging economy, has witnessed the rapid development of science over the past decades [3]. The Global Research Report by Web of Science (2019) states that in South and East Asia, Vietnam has the fastest growth rate with its indexed publication volume more than five-fold since 2009 and shows every sign of continuing expansion [2]. Despite this seemingly positive information, international collaboration in fact accounted for approximately 77% of all the publications [4,5].  Noticeably, the majority of these international projects were led by foreign authors [4]. From another perspective, studies utilizing network analysis to analyze co-authorship patterns among Vietnamese researchers indicate some signals of unsustainability. These studies found that in any given scientific group, most co-authors are centered around well-connected and highly productive researchers [5] and there are some indicators of unsustainable co-authorship groups due to their big gaps in productivity and connection [6].

In the regional context, among East and Southeast Asian countries, Vietnam still has low research intensity primarily due to its heavy reliance on foreign co-authors and limited quantity of international publications in applied and multidisciplinary fields [7]. During the 1991 – 2010 period, Vietnam researchers’ productivity is only equivalent to 13.33% of Singapore’s and 29% of Thailand’s [8,9].

Regarding the particularities of SSH, we have also explained the situation in the Introduction:

Among various disciplines, social sciences and humanities (SSH) appear to have a humble position compared with other fields in Vietnam and have been disparaged for their low productivity in both the total number and the adjusted number in comparison with natural science and technology disciplines [3,10].  By definition, SSH is the study of human behavior and interaction in social, cultural, environmental, economic, and political contexts [11]. Currently, SSH is not only implemented in the context of each separate discipline; instead, contemporary SSH projects are inherently interdisciplinary, taking place in different scientific and organizational environments [12]. Compared to natural sciences, SSH is more local-oriented [13]. Therefore, in the context of Vietnam’s fast-growing economy, SSH plays an important role in investigating the sociocultural as well as economic situations to provide reliable implications for policy makers as well as practitioners. 

  • Such issues remain opaque, unfortunately, for the numbers are presented as such. Besides the data provided, the authors engage quite uncritically with it. It would make the submission much stronger if the authors track potential downsides on the sheer quantitative approach especially in regard to Western-indexed journals, the JIF, and the particularities of the analysed disciplines. I go into more detail about these issues in the following points. Some of these, especially the terminological corrections, seem necessary for publication. The latter points about potential inclusions of relative numbers and lines of discussion would strengthen the paper, but may not be necessary as such; they are more proposals to consider, which I hope are helpful.

Thank you for your insightful suggestion. We have acknowledged the downside of our approach in the Limitations and Future Research Direction section:

In this section, we would like to acknowledge several limitations of this article. Firstly, we approach the subject of internal research capacity from a sheer quantitative approach, especially focusing on ISI/Scopus-indexed journals. The journals, books or conference proceedings that are indexed in ISI or Scopus database are potentially biased toward English language and Western-oriented publications. Even though the usage of these databases is common practice [4,5], the lack of scientific publications in other languages and journals might hinder the true capacity of Vietnamese SSH researchers. Moreover, social sciences and humanities are qualitative and hermeneutic disciplines. Therefore, the usage of quantitative approach and counting publications only present an initial effort to provide a perspective regarding Vietnamese SSH. In future research studies, different approaches should be used to discuss the issue more critically.

  • What disciplines are subsumed under SSH. Did this change over the time the data represents?

The SSHPA system collects paper from 31 disciplines in SSH research. Detailed information of these disciplines was listed in Table 1. Disciplines in Social Sciences and Humanities in Vietnam (2008 – 2019):

The SSHPA system collects paper from 31 fields in SSH research. Detailed information of these disciplines was listed in Table 1.

Table 1: Disciplines in Social Sciences and Humanities in Vietnam (2008 – 2019)

Field

Number of Articles

Number of Authors

Economics

941

1350

Education

484

534

Health Care

451

967

Business

362

580

Sociology

213

435

Environment/Sustainability Science

202

538

Management

154

269

Agriculture

143

326

Law

95

79

Tourism

90

121

Political Science

84

115

Psychology

63

189

Media/Journalism

40

53

Linguistics

39

33

International Relations

38

31

Philosophy

30

74

Cultural Studies

30

66

Urban Studies

27

55

Anthropology

24

44

Applied Math

23

39

Scientometrics

18

43

Geography

18

57

Forestry

15

33

Demography

15

32

History

13

21

Art

11

24

Logistics

10

18

Asian Studies

7

16

Archeology

7

36

Literature

6

3

Statistics

4

11

Architecture

3

16

  • 2: “only 740 (12.2%) of the 6,031 cited articles” > Could you relate this to the proportion of researchers in Vietnam? 12% does seem low intuitively. But maybe the articles are exceptional in themselves and so the low number is made up for with quality. Moreover, can you ascertain whether there are more than 12% researchers in SSH disciplines in Vietnam so that the authorship number indeed deviates in relation?

This information have been clarified in the Literature Review section:

Regarding research output, Vietnam’s research performance has achieved remarkable success in raising scientific productivity [8]. In particular, the Global Research Report by Web of Science (2019) states that in South and East Asia, Vietnam has the fastest growth rate with its indexed publication volume more than five-fold since 2009 and shows every sign of continuing expansion [2]. According to this report, Vietnam has 294 researchers per million population, which ranks fourth only after Singapore, Malaysia and Thailand. The annually average publication output on Web of Science from 2014 to 2018 is 3,766 papers, which ranks sixth in the region after India, Singapore, Malaysia, Pakistan and Thailand. However, impact profiles for papers published during 2009 – 2018 also show that amongst more than 26,700 papers of Vietnam, 78% of them are conducted with international collaboration [2].

In addition, regarding research quality, the report presents a widely used indicator namely the Category-Normalized Citation Impact (CNCI), which reflects a publication’s academic impact of a country. The world average is used as a reference benchmark at 1.0. Notably, despite the positive signal about Vietnam’s scientific output, CNCI has shown some large disparity between the domestic scientific impact (CNCI domestic) which removes all publications with international co-authorship and the gross one (CNCI gross) which counts all works with at least one national address. In particular, the CNCI gross, at 1.20, is almost twice as much as the CNCI domestic at 0.67, which indicates the limited internal capabilities of Vietnamese researchers [2].

[…]

In a previous work, data from a sample of 412 Vietnamese scholars who have published in Scopus-indexed journals between 2008 and 2017 revealed that more than 90% of social scientists collaborated with other authors to publish articles with an average of 13 times [28]. The data system also showed the prevalence of international cooperation between 1996 and 2013, accounting for 77% of the growth, which proved that Vietnamese scientific outputs depend heavily on foreign authors [4,29]. By analyzing the network of 412 authors, Ho, T.M et. al found that the network level of connection was low with only 0.47%, while the clustering factor was very high (58.64%) [30]. It was a demonstration that the communication and knowledge exchange between Vietnamese scholars was inefficient and socially unsustainable [31,32].

  • 2: NAFOSTED was established to “incentivize researchers” to do what? Is publishing more an incentive and measure of success?

National Foundation for Science and Technology (NAFOSTED) is the Vietnamese equivalence of United States’ National Science Foundation (NSF):

As an initiative to promote scientific works, from 2008, the first national scientific funding agency – the National Foundation for Science and Technology (NAFOSTED), was established by the Vietnamese government in order to financially support in both natural and social science fields. Together with the later issuing of Circular no. 23/2014/ND-CP, this has spurred on progress towards making this foundation-funded scientific projects compliant to international standards. Until 2019, there have been more than 10,000 scientists participating in the research, over 2,400 trained doctors, and over 4,000 ISI articles were published through nearly 2,800 research projects funded and supported by NAFOSTED sponsorship [27].

  • Figure 2: This is an interesting representation. Yet, you do not provide an account for the numbers behind it as well as a legend clarifying what the double connection (double lines between sites of knowledge production) stands for. It would be good to have more information about this, especially what you refer to with “national collaboration network”. Are the researchers just collaborating more or are there new researchers and new institutions which initiates the increase by default?

We have provided Figure 4. Number of affiliation and article in domestic collaboration during 2008-2019 (Last updated 30 April, 2020) to contextualize Figure 3 (previously 2).

  • The description of figures in table 2 is a bit unclear in that the terms “author” and “researcher” seem to be mixed up.

The description of figures in Table 2 is also edited for clarification. However, it should be noted that we used “author” and “researcher” interchangeably in this article:

From 2008 to 2019, there were 1369 publications written by Vietnamese authors only and 1810 internationally collaborated publications. It is notable that within three years from 2017 to 2019, the percentage of papers authored exclusively by Vietnamese rose by a significant amount and even surpassed the percentage of international collaborated papers the first time in 2019 with 55.83% and 44.17% respectively (see Table 3). This shows an improvement in the internal capability of Vietnam SSH.

  • “among the total 6,401 researchers during the period 2008-2019” > This total denotates the total of authors, correct?

This is our mistake. We have deleted the sentence. Thank you.

  • The description says “female researchers accounted for” while the table says “female authors”. I assume the latter is correct. If so, could you rectify this in the text?

It should be noted that we used “author” and “researcher” interchangeably in this article.

  • What is the nominative here? I.e. what can the numbers tell us beyond mere authorship, is there are relative measure?

We have used the average percentage number to tell beyond mere authorship. Moreover, we have also provided Figure 5. Number of “new” Vietnamese authors participating in SS&H research by gender in the period 2008 - 2019 (Last updated 30 April, 2020) to further illustrate the issue.

  • Is it possible to sort out repeated authorship for these numbers?

It is possible to sort our repeated authorship. However, we believe the number of “new” Vietnamese authors is more meaningful, as it records those who has their first publication in the system. Thus, there is no repeated authorship in the number of “new” authors.

  • Again here, after clarifying on the terminology, it would be good to relate the two categories (institutionalised researchers and authors), if possible.

It should be noted that we used “author” and “researcher” interchangeably in this article.

  • Table 3: Can you field-normalise these numbers? The JIF as such should not be used as a general metric of quality. If at all, it should only be used as a comparative metric within a field, looking at generalisable citation patterns. You objectify the quality of individual authors by means of generalised JIF numbers, which seems inappropriate.

We have deleted the JIF table due to the problem with this measurement.

  • Meaning of the numbers raised: For instance in regard to gender, the reader cannot infer too much meaning since gender among authorship is something different than gender among researchers in Vietnam. As SSH are qualitative and hermeneutic disciplines, it may quite as well be the case that crude more of publications is not necessarily good, so inferring a critique based on authorship numbers alone may be short-sighted.

We have acknowledged this issue in the Limitations:

In this section, we would like to acknowledge several limitations of this article. Firstly, we approach the subject of internal research capacity from a sheer quantitative approach, especially focusing on ISI/Scopus-indexed journals. The journals, books or conference proceedings that are indexed in ISI or Scopus database are potentially biased toward English language and Western-oriented publications. Even though the usage of these databases is common practice [4,5]  the lack of scientific publications in other languages and journals might hinder the true capacity of Vietnamese SSH researchers. Moreover, social sciences and humanities are qualitative and hermeneutic disciplines. Therefore, the usage of quantitative approach and counting publications only present an initial effort to provide a perspective regarding Vietnamese SSH. In future research studies, different approaches should be used to discuss the issue more critically. 

  • In their current single representation, the numbers are not quite relatable, neither normatively regarding native researcher growth, nor relatively regarding other clusters of disciplines. It wold be good, therefore, to include such comparisons in the study. If such comparisons are not available, you can hardly infer particularities of authorship in the SSH fields.
  • If there are only authorship numbers available and no numbers of researcher and institutional growth to relate these to, it would be good to at least provide comparisons to other disciplines.

We have provided Figure 2 to discuss the number in each SSH fields:

In particular, the discipline with the largest number of authors is Economics, which also has a rather balance in the types of collaboration (581 authors collaborate domestically, and 596 authors publishing with international authors). Health care is the field with the most significant difference (594 internationally collaborated authors; and only 138 domestically collaborated authors). Out of 30 disciplines, there are 17 fields in which Vietnamese authors mainly works with domestic authors, typically Demography, Economics, Asian Studies, Law...

  • On page 7, you discuss the scholarly success and “collaboration is an effective way to disseminate knowledge.” And yet, it is questionable what the role of authorship in these indexed journals is. Since the indices you mention are usually attributed a Western bias, do the journals also have such bias? If so, what does this mean for scholarship in Vietnam? What role has international authorship visibility in the national context of developing research policies. This may be an important discussion to be included since you base your analysis on authorship alone.

We have discussed this issue in the Discussion:

Domestic SSH researchers have been asserting their capacity to grow in terms of output and playing the lead author role. Out of 3,131 publications, 77.26% was led by Vietnamese author. Moreover, the ratio of Vietnamese lead authors to foreign lead authors has risen by more than three-fold, from 2.06 in 2008 to 6.69 in 2019. The number of solo authors also has an increasing trend. These results highlight the growing independence of Vietnamese SSH researchers from collaborations with foreign authors, especially in 2017-2019 with a drop in the number of foreign lead authors compared to 2016. The high number of foreign authors in the 2008-2016 period might be an impact of the focus on sending students to study oversea for a long period [34]. Those who have studied oversea might published their first papers with their advisors and colleagues oversea. Then, only until recently that they have started to move back to Vietnam to train a new generation of researchers. As the leading researchers are trained in the Western institutions, the promotion of international scientific publications in ISI/Scopus-indexed journals are inevitable. The proactive attitude of Vietnamese government through the Circular 08/2017/TT-BGDDT, which outlines the regulations on doctoral enrolment and training might have factored into the promising development of the Vietnamese academic community [35-37]. Moreover, as the Vietnamese universities are making changes to upgrade the quality, the usage of international ranking boards such as Times Higher Education or QS is common. The reliance on the criteria of these ranking boards also contribute to the focus on scientific publications.

  • Furthermore, in the same line of discussion may be a perspective on analysing authorship as a placeholder for scholarly achievement. Since you are working on SSH scholarship, what could go wrong if policies incentivise sheer higher numbers of publications irrespective of correspondingly developing research programmes? Why is this particularly a danger for scholarship in SSH disciplines? What may be dangers of internationalising authorship particularly in humanities disciplines for they are usually rooted in regional scholarship?

We have discussed this issue in the Discussion:

From another aspect, the act of promoting higher number of publications by incentivizing researchers to achieve some kinds of “key performance indicators” might also have several consequences. It should be noted that within an emerging economy context with limited budget for science as a whole, the financial resource should be allocated not only for rewarding high-performers but also to develop training programs for early-career researchers, which is also a way to sustain the high-quality human resource. The promotion of scientific productivity, which leads to an intense level of international collaboration, may also result in some dangers for SSH disciplines, as the research in these fields are usually rooted in the local contexts. Therefore, cooperating with foreign authors, or even being led by these colleagues may lead to a higher chance of incorporating modern theories and frameworks, most of which from the Western world. This means that the diversity and originality of the research may be compromised.

  • There are some language issues, especially regarding word choice and sentence structure.

We have revised the paper thoroughly to make sure there is no language issue left.

  • Many thanks for the paper and the request to have me review it!

Thank you for your insightful comments. You effort have helped us improving our article significantly.

We believe that we have put an appropriate amount of efforts in responding to all of your concerns. We hope that you find our revision satisfactory, and would like to once again thank you for your suggestions. We are truly honored to be able to work with you in improving this manuscript.

With our highest respects,

The authors

Round 2

Reviewer 2 Report

First of all, the indications and responses above must be seen in the light of the following repairs.

Apparently the article in its second verson has incorporated some of the indications given. However, the continuity of the review depends on some indispensable corrections. Without them, reading and understanding this new text is not possible.

  1. the authors added new figures and new tables.

The sequence of figures and tables has no logic and doen’t enable an enchained and clear comprehension

Figure 1 appears on page 7.

The next figure is Figure 6, page 8.

The last figure is Figure 4, page 12.

 7 figures are presented. There is no consistency.

The same for Tables:

Table 1 appears on page 5.

The next table is Table 4, which appears on page 8.

The following table is Table 2, page 10

There is no coherence, again.

2.English language, mainly in new pieces of text, should be corrected. Some examples of what needs to be reviewed

page 4 - "To expand the findings of previous studies by examining the collaboration patterns of authors in Vietnam’s SSH both domestically and internationally". Something is missing.

page 5 - finished phrases "in": "The system’s creation process and logical structure can be found in [29]". This isn't correct.

Author Response

Letter of detailed responses to Reviewer 2

Dear Sir/Madam, 

Thank you very much for your time and efforts in reviewing our manuscript. Your detailed comments have helped us a lot to improve the quality of our paper.  

We have addressed your points in our revised version. Please notice that in the revised paper, the parts that are highlighted in yellow is for correction on the old text, the parts highlighted in green is written anew. Below are our answers to your comments (in italic). Also, the line numbers in the text refer to the revised paper.

First of all, the indications and responses above must be seen in the light of the following repairs.

Apparently the article in its second verson has incorporated some of the indications given. However, the continuity of the review depends on some indispensable corrections. Without them, reading and understanding this new text is not possible.

the authors added new figures and new tables.

The sequence of figures and tables has no logic and doen’t enable an enchained and clear comprehension

Figure 1 appears on page 7.

The next figure is Figure 6, page 8.

The last figure is Figure 4, page 12.

 7 figures are presented. There is no consistency.

The same for Tables:

Table 1 appears on page 5.

The next table is Table 4, which appears on page 8.

The following table is Table 2, page 10

There is no coherence, again.

Thank you for pointing out this silly mistake. We have corrected numbering of figures and tables to maintain the coherence of the article. We hope that the flow of the article is now easier to follow. In detail:

Table 1. The numbering sequence of tables

Table

Caption

Page

Table 1:

Disciplines in Social Sciences and Humanities in Vietnam (2008 – 2019)

5

Table 2:

Number of authors participating in Social Sciences and Humanities research by gender during 2008-2019 (Last updated 30 April 2020)

8

Table 3:

Top 20 countries in Vietnam research collaboration network during 2008 - 2019 (Last updated 30 April 2020)

10

Table 4:

Articles whose authors are all Vietnamese vs. Articles that have non-Vietnamese authors during 2008-2019 (Last updated 30 April 2020)

11

Table 2. The numbering sequence of figures

Figure

Caption

Page

Figure 1:

The number of Vietnamese and foreign lead authors in SSH research during 2008-2019 (Last updated 30 April 2020)

7

Figure 2:

Number of Vietnamese lead author by gender from 2008 to 2019 (Last updated 30 April 2020)

8

Figure 3:

Number of “new” Vietnamese authors participating in SS&H research by gender in the period 2008 - 2019 (Last updated 30 April 2020)

9

Figure 4:

Number of solo authors by gender from 2008 to 2019 (Last updated 30 April 2020)

10

Figure 5:

The number of publications domestically collaborated and internationally collaborated by discipline during 2008-2019 (Last updated 30 April 2020)

12

Figure 6:

A map of Vietnam domestic collaboration a) 2008 and b) 2019 (Last updated 30 April 2020)

13

Figure 7:

Number of affiliation and article in domestic collaboration during 2008-2019 (Last updated 30 April, 2020)

13

2.English language, mainly in new pieces of text, should be corrected. Some examples of what needs to be reviewed

page 4 - "To expand the findings of previous studies by examining the collaboration patterns of authors in Vietnam’s SSH both domestically and internationally". Something is missing.

page 5 - finished phrases "in": "The system’s creation process and logical structure can be found in [29]". This isn't correct.

According to your suggestion, we have proofread the article carefully to make sure the English is comprehensible and contains no grammatical errors. For example, in Page 4, the sentence is now changed to “Additionally, the article aims to expand the findings of previous studies by examining the collaboration patterns of authors in Vietnam’s SSH both domestically and internationally.” Furthermore, in Page 5, the sentence is now changed to “The system’s creation process and logical structure can be found in detail in the article by Vuong et al. (2018) [29].”

Please see the details of revision in the manuscript.

In closing, we highly appreciate the hard work and time that you have spent on this manuscript. Your comments have helped us a lot in improving the quality of our paper. We hope that the revised paper has met your requirements.  

Please accept our sincere thanks for your great contributions to the improvement of financial studies and the overall advancement of sciences in the world. 

Shall you have further comments, we look forward to hearing from you.  

Best regards, 

Reviewer 3 Report

Thanks a lot for your clarifications. You have put a lot of effort into expressing in more detail the contours of your data and potential shortcomings of the analysis. You also contextualise the data in their empirical reality. I think this makes it much stronger for readers to learn from your study.
The additional figures further highlight this. However, the numbering of tables and figures has gone awry: it jumps around from figure 1 to 6 to 5 to 7 back to 2 and so on. I can imagine it be a tedious task to sort this out, but this will be necessary for the reader to follow the argument properly. Other than this, I have no further remarks.

Thank you again for your study. It will be good for the international context to learn more about regional variations and change such as you present them!

Author Response

Letter of detailed responses to Reviewer 3

Dear Sir/Madam, 

Thank you very much for your time and efforts in reviewing our manuscript. Your detailed comments have helped us a lot to improve the quality of our paper.  

We have addressed your points in our revised version. Please notice that in the revised paper, the parts that are highlighted in yellow is for correction on the old text, the parts highlighted in green is written anew. Below are our answers to your comments (in italic). Also, the line numbers in the text refer to the revised paper.

Thanks a lot for your clarifications. You have put a lot of effort into expressing in more detail the contours of your data and potential shortcomings of the analysis. You also contextualise the data in their empirical reality. I think this makes it much stronger for readers to learn from your study.

Thank you for your encouraging comments. We hope that the article will provide insightful information to the audience.

The additional figures further highlight this. However, the numbering of tables and figures has gone awry: it jumps around from figure 1 to 6 to 5 to 7 back to 2 and so on. I can imagine it be a tedious task to sort this out, but this will be necessary for the reader to follow the argument properly. Other than this, I have no further remarks.

Thank you for pointing out this silly mistake. We have revised the numbering of the tables and figures to make sure the flow of the article is easy to read. In detail:

Table 1. The numbering sequence of tables

Table

Caption

Page

Table 1:

Disciplines in Social Sciences and Humanities in Vietnam (2008 – 2019)

5

Table 2:

Number of authors participating in Social Sciences and Humanities research by gender during 2008-2019 (Last updated 30 April 2020)

8

Table 3:

Top 20 countries in Vietnam research collaboration network during 2008 - 2019 (Last updated 30 April 2020)

10

Table 4:

Articles whose authors are all Vietnamese vs. Articles that have non-Vietnamese authors during 2008-2019 (Last updated 30 April 2020)

11

Table 2. The numbering sequence of figures

Figure

Caption

Page

Figure 1:

The number of Vietnamese and foreign lead authors in SSH research during 2008-2019 (Last updated 30 April 2020)

7

Figure 2:

Number of Vietnamese lead author by gender from 2008 to 2019 (Last updated 30 April 2020)

8

Figure 3:

Number of “new” Vietnamese authors participating in SS&H research by gender in the period 2008 - 2019 (Last updated 30 April 2020)

9

Figure 4:

Number of solo authors by gender from 2008 to 2019 (Last updated 30 April 2020)

10

Figure 5:

The number of publications domestically collaborated and internationally collaborated by discipline during 2008-2019 (Last updated 30 April 2020)

12

Figure 6:

A map of Vietnam domestic collaboration a) 2008 and b) 2019 (Last updated 30 April 2020)

13

Figure 7:

Number of affiliation and article in domestic collaboration during 2008-2019 (Last updated 30 April, 2020)

13

Thank you again for your study. It will be good for the international context to learn more about regional variations and change such as you present them!

We highly appreciate the hard work and time that you have spent on this manuscript. Your comments have helped us a lot in improving the quality of our paper. We hope that the revised paper has met your requirements.  

Please accept our sincere thanks for your great contributions to the improvement of financial studies and the overall advancement of sciences in the world. 

Shall you have further comments, we look forward to hearing from you.  

Best regards, 

Round 3

Reviewer 2 Report

  1. the content of 6. Limitations should appear in the section related to Materials and methods
  2. English language changes required. "to demonstrates"? Before returning a new version check all the text
  3. What is the difference between "new" author and new author? What is a new author?
  4. Is it necessary to separate 3.2 and 3.3?
  5. Abstract should mention the quantiative approach of the article and its role as "an initial effort to..."